# A Unified Game-Theoretic Approach to Multiagent Reinforcement Learning

**Marc Lanctot**
DeepMind
lanctot@

**Vinicius Zambaldi**
DeepMind
vzambaldi@

**Audrūnas Gruslys**
DeepMind
audrunas@

**Angeliki Lazaridou**
DeepMind
angeliki@

**Karl Tuyls**
DeepMind
karltuyls@

**Julien Pérolat**
DeepMind
perolat@

**David Silver**
DeepMind
davidsilver@

**Thore Graepel**
DeepMind
thore@

...@google.com

## Abstract

To achieve general intelligence, agents must learn how to interact with others in a shared environment: this is the challenge of multiagent reinforcement learning (MARL). The simplest form is *independent reinforcement learning* (InRL), where each agent treats its experience as part of its (non-stationary) environment. In this paper, we first observe that policies learned using InRL can overfit to the other agents' policies during training, failing to sufficiently generalize during execution. We introduce a new metric, *joint-policy correlation*, to quantify this effect. We describe an algorithm for general MARL, based on approximate best responses to mixtures of policies generated using deep reinforcement learning, and empirical game-theoretic analysis to compute meta-strategies for policy selection. The algorithm generalizes previous ones such as InRL, iterated best response, double oracle, and fictitious play. Then, we present a scalable implementation which reduces the memory requirement using decoupled meta-solvers. Finally, we demonstrate the generality of the resulting policies in two partially observable settings: gridworld coordination games and poker.

## 1 Introduction

Deep reinforcement learning combines deep learning [59] with reinforcement learning [94, 64] to compute a policy used to drive decision-making [73, 72]. Traditionally, a single agent interacts with its environment repeatedly, iteratively improving its policy by learning from its observations. Inspired by recent success in Deep RL, we are now seeing a renewed interest in *multiagent* reinforcement learning (MARL) [90, 17, 99]. In MARL, several agents interact and learn in an environment simultaneously, either competitively such as in Go [91] and Poker [39, 105, 74], cooperatively such as when learning to communicate [23, 93, 36], or some mix of the two [60, 95, 35].

The simplest form of MARL is *independent RL* (InRL), where each learner is oblivious to the other agents and simply treats all the interaction as part of its ("localized") environment. Aside from the problem that these local environments are non-stationary and non-Markovian [57] resulting in a loss of convergence guarantees for many algorithms, the policies found can overfit to the other agents' policies and hence not generalize well. There has been relatively little work done in RL community on overfitting to the environment [102, 69], but we argue that this is particularly important in multiagent settings where one must react dynamically based on the observed behavior of others. Classical techniques collect or approximate extra information such as the joint values [62, 19, 29, 56],

use adaptive learning rates [12], adjust the frequencies of updates [48, 81], or dynamically respond to the other agents actions online [63, 50]. However, with the notable exceptions of very recent work [22, 80], they have focused on (repeated) matrix games and/or the fully-observable case.

There have several proposals for treating partial observability in the multiagent setting. When the model is fully known and the setting is strictly adversarial with two players, there are policy iteration methods based on regret minimization that scale very well when using domain-specific abstractions [27, 14, 46, 47], which was a major component of the expert no-limit poker AI Libratus [15]; recently these methods were combined with deep learning to create an expert no-limit poker AI called DeepStack [74]. There is a significant amount of work that deals with the case of decentralized cooperative problems [76, 79], and in the general setting by extending the notion of belief states and Bayesian updating from POMDPs [28]. These models are quite expressive, and the resulting algorithms are fairly complex. In practice, researchers often resort to approximate forms, by sampling or exploiting structure, to ensure good performance due to intractability [41, 2, 68].

In this paper, we introduce a new metric for quantifying the correlation effects of policies learned by independent learners, and demonstrate the severity of the overfitting problem. These coordination problems have been well-studied in the fully-observable cooperative case [70]: we observe similar problems in a partially-observed mixed cooperative/competitive setting and, and we show that the severity increases as the environment becomes more partially-observed. We propose a new algorithm based on economic reasoning [82], which uses (i) deep reinforcement learning to compute best responses to a distribution over policies, and (ii) empirical game-theoretic analysis to compute new meta-strategy distributions. As is common in the MARL setting, we assume centralized training for decentralized execution: policies are represented as separate neural networks and there is no sharing of gradients nor architectures among agents. The basic form uses a centralized payoff table, which is removed in the distributed, decentralized form that requires less space.

## 2   Background and Related Work

In this section, we start with basic building blocks necessary to describe the algorithm. We interleave this with the most relevant previous work for our setting. Several components of the general idea have been (re)discovered many times across different research communities, each with slightly different but similar motivations. One aim here is therefore to unify the algorithms and terminology.

A **normal-form game** is a tuple $(\Pi, U, n)$ where $n$ is the number of players, $\Pi = (\Pi_1, \cdots, \Pi_n)$ is the set of policies (or strategies, one for each player $i \in [[n]]$, where $[[n]] = \{1, \cdots, n\}$), and $U : \Pi \to \Re^n$ is a payoff table of utilities for each joint policy played by all players. **Extensive-form games** extend these formalisms to the multistep sequential case (*e.g.* poker).

Players try to maximize their own expected utility. Each player does this by choosing a policy from $\Pi_i$, or by sampling from a mixture (distribution) over them $\sigma_i \in \Delta(\Pi_i)$. In this multiagent setting, the quality of $\sigma_i$ depends on other players' strategies, and so it cannot be found nor assessed independently. Every finite extensive-form game has an equivalent normal-form [53], but since it is exponentially larger, most algorithms have to be adapted to handle the sequential setting directly.

There are several algorithms for computing strategies. In zero-sum games (where $\forall \pi \in \Pi, \vec{1} \cdot U(\pi) = 0$), one can use e.g. linear programming, fictitious play [13], replicator dynamics [97], or regret minimization [8]. Some of these techniques have been extended to extensive (sequential) form [39, 25, 54, 107] with an exponential increase in the size of the state space. However, these extensions have almost exclusively treated the two-player case, with some notable exceptions [54, 26]. Fictitious play also converges in potential games which includes cooperative (identical payoff) games.

The **double oracle** (DO) algorithm [71] solves a set of (two-player, normal-form) subgames induced by subsets $\Pi^t \subset \Pi$ at time $t$. A payoff matrix for the subgame $G_t$ includes only those entries corresponding to the strategies in $\Pi^t$. At each time step $t$, an equilibrium $\sigma^{*,t}$ is obtained for $G^t$, and to obtain $G^{t+1}$ each player adds a best response $\pi_i^{t+1} \in BR(\sigma_{-i}^{*,t})$ from the full space $\Pi_i$, so for all $i$, $\Pi_i^{t+1} = \Pi_i^t \cup \{\pi_i^{t+1}\}$. The algorithm is illustrated in Figure 1. Note that finding an equilibrium in a zero-sum game takes time polynomial in $|\Pi^t|$, and is PPAD-complete for general-sum [89].

Clearly, DO is guaranteed to converge to an equilibrium in two-player games. But, in the worst-case, the entire strategy space may have to be enumerated. For example, this is necessary for Rock-Paper-

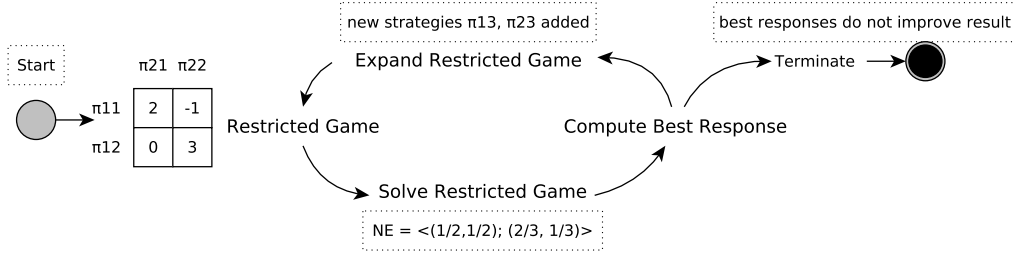

Figure 1: The Double Oracle Algorithm. Figure taken from [10] with authors' permission.

Scissors, whose only equilibrium has full support $\left(\frac{1}{3}, \frac{1}{3}, \frac{1}{3}\right)$. However, there is evidence that support sizes shrink for many games as a function of episode length, how much hidden information is revealed and/or affects it has on the payoff [65, 86, 10]. Extensions to the extensive-form games have been developed [67, 9, 10] but still large state spaces are problematic due to the curse of dimensionality.

Empirical game-theoretic analysis (EGTA) is the study of meta-strategies obtained through simulation in complex games [100, 101]. An **empirical game**, much smaller in size than the full game, is constructed by discovering strategies, and meta-reasoning about the strategies to navigate the strategy space. This is necessary when it is prohibitively expensive to explicitly enumerate the game's strategies. Expected utilities for each joint strategy are estimated and recorded in an **empirical payoff table**. The empirical game is analyzed, and the simulation process continues. EGTA has been employed in trading agent competitions (TAC) and automated bidding auctions.

One study used evolutionary dynamics in the space of known expert meta-strategies in Poker [83]. Recently, reinforcement learning has been used to *validate* strategies found via EGTA [104]. In this work, we aim to discover new strategies through learning. However, instead of computing exact best responses, we compute approximate best responses using reinforcement learning. A few epochs of this was demonstrated in continuous double auctions using tile coding [87]. This work follows up in this line, running more epochs, using modern function approximators (deep networks), a scalable implementation, and with a focus on finding policies that can generalize across contexts.

A key development in recent years is deep learning [59]. While most work in deep learning has focused on supervised learning, impressive results have recently been shown using deep neural networks for reinforcement learning, e.g. [91, 38, 73, 77]. For instance, Mnih et al. [73] train policies for playing Atari video games and 3D navigation [72], given only screenshots. Silver et al. introduced AlphaGo [91, 92], combining deep RL with Monte Carlo tree search, outperforming human experts.

Computing approximate responses is more computationally feasible, and fictitious play can handle approximations [42, 61]. It is also more biologically plausible given natural constraints of bounded rationality. In **behavioral game theory** [103], the focus is to *predict* actions taken by humans, and the responses are intentionally constrained to increase predictive ability. A recent work uses a deep learning architecture [34]. The work that closely resembles ours is **level-k** thinking [20] where level $k$ agents respond to level $k - 1$ agents, and more closely **cognitive hierarchy** [18], in which responses are to distributions over levels $\{0, 1, \ldots, k - 1\}$. However, our goals and motivations are very different: we use the setup as a means to produce more general policies rather than to predict human behavior. Furthermore, we consider the sequential setting rather than normal-form games.

Lastly, there has been several studies from the literature on co-evolutionary algorithms; specifically, how learning cycles and overfitting to the current populations can be mitigated [78, 85, 52].

## 3    Policy-Space Response Oracles

We now present our main conceptual algorithm, policy-space response oracles (PSRO). The algorithm is a natural generalization of Double Oracle where the meta-game's choices are policies rather than actions. It also generalizes Fictitious Self-Play [39, 40]. Unlike previous work, any meta-solver can be plugged in to compute a new meta-strategy. In practice, parameterized policies (function approximators) are used to generalize across the state space without requiring any domain knowledge.

The process is summarized in Algorithm 1. The meta-game is represented as an empirical game, starting with a single policy (uniform random) and growing, each epoch, by adding policies ("oracles")

| **Algorithm 1:** Policy-Space Response Oracles | **Algorithm 2:** Deep Cognitive Hierarchies |
|---|---|
| **input** :initial policy sets for all players $\Pi$ | **input** :player number $i$, level $k$ |
| Compute exp. utilities $U^\Pi$ for each joint $\pi \in \Pi$ | **while** *not terminated* **do** |
| Initialize meta-strategies $\sigma_i = \text{UNIFORM}(\Pi_i)$ | $\quad$ CHECKLOADMS($\{j|j \in [[n]], j \neq i\}, k$) |
| **while** *epoch e in $\{1, 2, \cdots\}$* **do** | $\quad$ CHECKLOADORACLES($j \in [[n]], k' \leq k$) |
| $\quad$ **for** *player $i \in [[n]]$* **do** | $\quad$ CHECKSAVEMS($\sigma_{i,k}$) |
| $\quad\quad$ **for** *many episodes* **do** | $\quad$ CHECKSAVEORACLE($\pi_{i,k}$) |
| $\quad\quad\quad$ Sample $\pi_{-i} \sim \sigma_{-i}$ | $\quad$ Sample $\pi_{-i} \sim \sigma_{-i,k}$ |
| $\quad\quad\quad$ Train oracle $\pi'_i$ over $\rho \sim (\pi'_i, \pi_{-i})$ | $\quad$ Train oracle $\pi_{i,k}$ over $\rho_1 \sim (\pi_{i,k}, \pi_{-i})$ |
| $\quad\quad$ $\Pi_i = \Pi_i \cup \{\pi'_i\}$ | $\quad$ **if** *iteration number* **mod** $T_{ms} = 0$ **then** |
| $\quad$ Compute missing entries in $U^\Pi$ from $\Pi$ | $\quad\quad$ Sample $\pi_i \sim \sigma_{i,k}$ |
| $\quad$ Compute a meta-strategy $\sigma$ from $U^\Pi$ | $\quad\quad$ Compute $u_i(\rho_2)$, where $\rho_2 \sim (\pi_i, \pi_{-i})$ |
| Output current solution strategy $\sigma_i$ for player $i$ | $\quad\quad$ Update stats for $\pi_i$ and update $\sigma_{i,k}$ |
| | Output $\sigma_{i,k}$ for player $i$ at level $k$ |

that approximate best responses to the meta-strategy of the other players. In (episodic) partially observable multiagent environments, when the other players are fixed the environment becomes Markovian and computing a best response reduces to solving a form of MDP [30]. Thus, any reinforcement learning algorithm can be used. We use deep neural networks due to the recent success in reinforcement learning. In each episode, one player is set to *oracle*(learning) mode to train $\pi'_i$, and a fixed policy is sampled from the opponents' meta-strategies ($\pi_{-i} \sim \sigma_{-i}$). At the end of the epoch, the new oracles are added to their policy sets $\Pi_i$, expected utilities for new policy combinations are computed via simulation and added to the empirical tensor $U^\Pi$, which takes time exponential in $|\Pi|$.

Define $\Pi^T = \Pi^{T-1} \cup \pi'$ as the policy space including the currently learning oracles, and $|\sigma_i| = |\Pi_i^T|$ for all $i \in [[n]]$. Iterated best response is an instance of PSRO with $\sigma_{-i} = (0, 0, \cdots, 1, 0)$. Similarly, Independent RL and fictitious play are instances of PSRO with $\sigma_{-i} = (0, 0, \cdots, 0, 1)$ and $\sigma_{-i} = (1/K, 1/K, \cdots, 1/K, 0)$, respectively, where $K = |\Pi_{-i}^{T-1}|$. Double Oracle is an instance of PSRO with $n = 2$ and $\sigma^T$ set to a Nash equilibrium profile of the meta-game $(\Pi^{T-1}, U^{\Pi^{T-1}})$.

An exciting question is what can happen with (non-fixed) meta-solvers outside this known space? Fictitious play is agnostic to the policies it is responding to; hence it can only sharpen the meta-strategy distribution by repeatedly generating the same best responses. On the other hand, responses to equilibrium strategies computed by Double Oracle will (i) overfit to a specific equilibrium in the $n$-player or general-sum case, and (ii) be unable to generalize to parts of the space not reached by any equilibrium strategy in the zero-sum case. Both of these are undesirable when computing general policies that should work well in any context. We try to balance these problems of overfitting with a compromise: meta-strategies with full support that force (mix in) $\gamma$ exploration over policy selection.

### 3.1 Meta-Strategy Solvers

A meta-strategy solver takes as input the empirical game $(\Pi, U^\Pi)$ and produces a meta-strategy $\sigma_i$ for each player $i$. We try three different solvers: regret-matching, Hedge, and projected replicator dynamics. These specific meta-solvers accumulate values for each policy ("arm") and an aggregate value based on all players' meta-strategies. We refer to $u_i(\sigma)$ as player $i$'s expected value given all players' meta-strategies and the current empirical payoff tensor $U^\Pi$ (computed via multiple tensor dot products.) Similarly, denote $u_i(\pi_{i,k}, \sigma_{-i})$ as the expected utility if player $i$ plays their $k^{th} \in [[K]] \cup \{0\}$ policy and the other players play with their meta-strategy $\sigma_{-i}$. Our strategies use an exploration parameter $\gamma$, leading to a lower bound of $\frac{\gamma}{K+1}$ on the probability of selecting any $\pi_{i,k}$.

The first two meta-solvers (Regret Matching and Hedge) are straight-forward applications of previous algorithms, so we defer the details to Appendix A.[1] Here, we introduce a new solver we call **projected replicator dynamics** (PRD). From Appendix A, when using the asymmetric replicator dynamics, *e.g.* with two players, where $U^\Pi = (\mathbf{A}, \mathbf{B})$, the change in probabilities for the $k^{th}$ component (*i.e.,* the policy $\pi_{i,k}$) of meta-strategies $(\sigma_1, \sigma_2) = (\mathbf{x}, \mathbf{y})$ are:

$$\frac{dx_k}{dt} = x_k[(\mathbf{Ay})_k - \mathbf{x}^T \mathbf{Ay}], \qquad \frac{dy_k}{dt} = y_k[(\mathbf{x}^T \mathbf{B})_k - \mathbf{x}^T \mathbf{By}],$$

To simulate the replicator dynamics in practice, discretized updates are simulated using a step-size of $\delta$. We add a projection operator $P(\cdot)$ to these equations that guarantees exploration: $\mathbf{x} \leftarrow P(\mathbf{x} + \delta\frac{d\mathbf{x}}{dt})$, $\mathbf{y} \leftarrow P(\mathbf{y} + \delta\frac{d\mathbf{y}}{dt})$, where $P(\mathbf{x}) = \mathrm{argmin}_{\mathbf{x}' \in \Delta_{\gamma}^{K+1}}\{||\mathbf{x}' - \mathbf{x}||\}$, if any $x_k < \gamma/(K+1)$ or $\mathbf{x}$ otherwise, and $\Delta_{\gamma}^{K+1} = \{\mathbf{x} \mid x_k \geq \frac{\gamma}{K+1}, \sum_k x_k = 1\}$ is the $\gamma$-exploratory simplex of size $K+1$. This enforces exploratory $\sigma_i(\pi_{i,k}) \geq \gamma/(K+1)$. The PRD approach can be understood as directing exploration in comparison to standard replicator dynamics approaches that contain isotropic diffusion or mutation terms (which assume undirected and unbiased evolution), for more details see [98].

## 3.2  Deep Cognitive Hierarchies

While the generality of PSRO is clear and appealing, the RL step can take a long time to converge to a good response. In complex environments, much of the basic behavior that was learned in one epoch may need to be relearned when starting again from scratch; also, it may be desirable to run many epochs to get oracle policies that can recursively reason through deeper levels of contingencies.

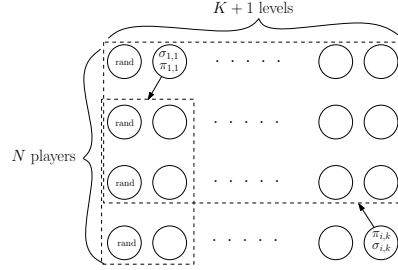

Figure 2: Overview of DCH

To overcome these problems, we introduce a practical parallel form of PSRO. Instead of an unbounded number of epochs, we choose a fixed number of *levels* in advance. Then, for an $n$-player game, we start $nK$ processes in parallel (level 0 agents are uniform random): each one trains a single oracle policy $\pi_{i,k}$ for player $i$ and level $k$ and updates its own meta-strategy $\sigma_{i,k}$, saving each to a central disk periodically. Each process also maintains copies of all the other oracle policies $\pi_{j,k' \leq k}$ at the current and lower levels, as well as the meta-strategies at the current level $\sigma_{-i,k}$, which are periodically refreshed from a central disk. We circumvent storing $U^{\Pi}$ explicitly by updating the meta-strategies online. We call this a Deep Cognitive Hierarchy (DCH), in reference to Camerer, Ho, & Chong's model augmented with deep RL. Example oracle response dynamics shown in Figure 2, and the pseudo-code in Algorithm 2.

Since each process uses slightly out-dated copies of the other process's policies and meta-strategies, DCH approximates PSRO. Specifically, it trades away accuracy of the correspondence to PSRO for practical efficiency and, in particular, scalability. Another benefit of DCH is an asymptotic reduction in total space complexity. In PSRO, for $K$ policies and $n$ players, the space required to store the empirical payoff tensor is $K^n$. Each process in DCH stores $nK$ policies of fixed size, and $n$ meta-strategies (and other tables) of size bounded by $k \leq K$. Therefore the total space required is $O(nK \cdot (nK + nK)) = O(n^2 K^2)$. This is possible is due to the use of *decoupled* meta-solvers, which compute strategies online without requiring a payoff tensor $U^{\Pi}$, which we describe now.

### 3.2.1  Decoupled Meta-Strategy Solvers

In the field of online learning, the experts algorithms ("full information" case) receive information about each arm at every round. In the bandit ("partial information") case, feedback is only given for the arm that was pulled. Decoupled meta-solvers are essentially sample-based adversarial bandits [16] applied to games. Empirical strategies are known to converge to Nash equilibria in certain classes of games (i.e. zero-sum, potential games) due to the folk theorem [8].

We try three: decoupled regret-matching [33], Exp3 (decoupled Hedge) [3], and decoupled PRD. Here again, we use exploratory strategies with $\gamma$ of the uniform strategy mixed in, which is also necessary to ensure that the estimates are unbiased. For decoupled PRD, we maintain running averages for the overall average value an value of each arm (policy). Unlike in PSRO, in the case of DCH, one sample is obtained at a time and the meta-strategy is updated periodically from online estimates.

## 4  Experiments

In all of our experiments, oracles use Reactor [31] for learning, which has achieved state-of-the-art results in Atari game-playing. Reactor uses Retrace($\lambda$) [75] for off-policy policy evaluation, and $\beta$-Leave-One-Out policy gradient for policy updates, and supports recurrent network training, which could be important in trying to match online experiences to those observed during training.

The action spaces for each player are identical, but the algorithms do not require this. Our implementation differs slightly from the conceptual descriptions in Section 3; see App. C for details.

**First-Person Gridworld Games.** Each agent has a local field-of-view (making the world partially observable), sees 17 spaces in front, 10 to either side, and 2 spaces behind. Consequently, observations are encoded as 21x20x3 RGB tensors with values $0 - 255$. Each agent has a choice of turning left or right, moving forward or backward, stepping left or right, not moving, or casting an endless light beam in their current direction. In addition, the agent has two composed actions of moving forward and turning. Actions are executed simultaneously, and order of resolution is randomized. Agents start on a random spawn point at the beginning of each episode. If an agent is touched ("tagged") by another agent's light beam twice, then the target agent is immediately teleported to a spawn point. In *laser tag*, the source agent then receives 1 point of reward for the tag. In another variant, *gathering*, there is no tagging but agents can collect apples, for 1 point per apple, which refresh at a fixed rate. In *pathfind*, there is no tagging nor apples, and both agents get 1 point reward when both reach their destinations, ending the episode. In every variant, an episode consists of 1000 steps of simulation. Other details, such as specific maps, can be found in Appendix D.

**Leduc Poker** is a common benchmark in Poker AI, consisting of a six-card deck: two suits with three cards (Jack, Queen, King) each. Each player antes 1 chip to play, and receives one private card. There are two rounds of betting, with a maximum of two raises each, whose values are 2 and 4 chips respectively. After the first round of betting, a single public card is revealed. The input is represented as in [40], which includes one-hot encodings of the private card, public card, and history of actions. Note that we use a more difficult version than in previous work; see Appendix D.1 for details.

## 4.1 Joint Policy Correlation in Independent Reinforcement Learning

To identify the effect of overfitting in independent reinforcement learners, we introduce **joint policy correlation** (JPC) matrices. To simplify the presentation, we describe here the special case of symmetric two-player games with non-negative rewards; for a general description, see Appendix B.2.

Values are obtained by running $D$ instances of the same experiment, differing only in the seed used to initialize the random number generators. Each experiment $d \in [[D]]$ (after many training episodes) produces policies $(\pi_1^d, \pi_2^d)$. The entries of each $D \times D$ matrix shows the mean return over $T = 100$ episodes, $\sum_{t=1}^{T} \frac{1}{T}(R_1^t + R_2^t)$, obtained when player 1 uses row policy $\pi_1^{d_i}$ and and player 2 uses column policy $\pi_2^{d_j}$. Hence, entries on the diagonals represent returns for policies that learned together (*i.e.,* same instance), while off-diagonals show returns from policies that trained in separate instances.

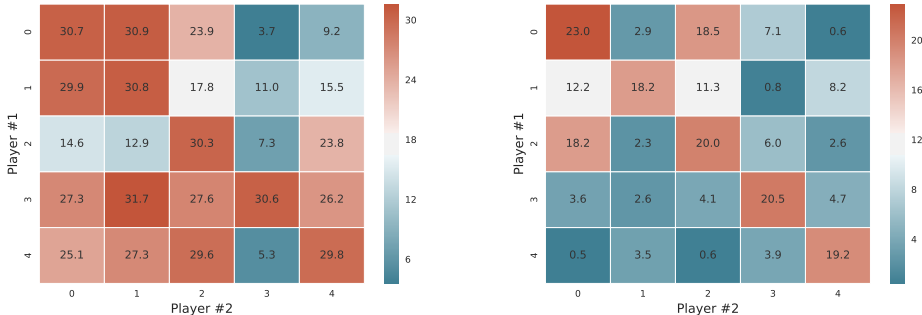

Figure 3: Example JPC matrices for InRL on Laser Tag small2 map (left) and small4 (right).

From a JPC matrix, we compute an **average proportional loss** in reward as $R_- = (\bar{D} - \bar{O})/\bar{D}$ where $\bar{D}$ is the mean value of the diagonals and $\bar{O}$ is the mean value of the off-diagonals. E.g. in Figure 3: $D = 30.44, O = 20.03, R_- = 0.342$. Even in a simple domain with almost full observability (small2), an independently-learned policy could expect to lose $34.2\%$ of its reward when playing with another independently-learned policy even though it *was trained under identical circumstances*! This clearly demonstrates an important problem with independent learners. In the other variants (gathering and pathfind), we observe no JPC problem, presumably because coordination is not required and the policies are independent. Results are summarized in Table 1. We have also noticed similar effects when using DQN [73] as the oracle training algorithm; see Appendix B.1 for example videos.

| Environment | Map | InRL | | | DCH(Reactor, 2, 10) | | | JPC Reduction |
|---|---|---|---|---|---|---|---|---|
| | | $\bar{D}$ | $\bar{O}$ | $R_-$ | $\bar{D}$ | $\bar{O}$ | $R_-$ | |
| Laser Tag | small2 | 30.44 | 20.03 | 0.342 | 28.20 | 26.63 | 0.055 | 28.7 % |
| Laser Tag | small3 | 23.06 | 9.06 | 0.625 | 27.00 | 23.45 | 0.082 | 54.3 % |
| Laser Tag | small4 | 20.15 | 5.71 | 0.717 | 18.72 | 15.90 | 0.150 | 56.7 % |
| Gathering | field | 147.34 | 146.89 | 0.003 | 139.70 | 138.74 | 0.007 | – |
| Pathfind | merge | 108.73 | 106.32 | 0.022 | 90.15 | 91.492 | < 0 | – |

Table 1: Summary of JPC results in first-person gridworld games.

We see that a (level 10) DCH agent reduces the JPC problem significantly. On small2, DCH reduces the expected loss down to 5.5%, 28.7% lower than independent learners. The problem gets larger as the map size grows and problem becomes more partially observed, up to a severe 71.7% average loss. The reduction achieved by DCH also grows from 28.7% to 56.7%.

**Is the Meta-Strategy Necessary During Execution?** The figures above represent the fully-mixed strategy $\sigma_{i,10}$. We also analyze JPC for only the highest-level policy $\pi_{i,10}$ in the laser tag levels. The values are larger here: $R_- = 0.147, 0.27, 0.118$ for small2-4 respectively, showing the importance of the meta-strategy. However, these are still significant reductions in JPC: 19.5%, 36.5%, 59.9%.

**How Many Levels?** On small4, we also compute values for level 5 and level 3: $R_- = 0.156$ and $R_- = 0.246$, corresponding to reductions in JPC of 56.1% and 44%, respectively. Level 5 reduces JPC by a similar amount as level 10 (56.1% vs 56.7%), while level 3 less so (44% vs. 56.1%.)

### 4.2 Learning to Safely Exploit and Indirectly Model Opponents in Leduc Poker

We now show results for a Leduc poker where strong benchmark algorithms exist, such as counter-factual regret (CFR) minimization [107, 11]. We evaluate our policies using two metrics: the first is performance against fixed players (random, CFR's average strategy after 500 iterations "cfr500", and a purified version of "cfr500pure" that chooses the action with highest probability.) The second is commonly used in poker AI: $\textsc{NashConv}(\sigma) = \sum_i^n \max_{\sigma_i' \in \Sigma_i} u_i(\sigma_i', \sigma_{-i}) - u_i(\sigma)$, representing how much can be gained by deviating to their best response (unilaterally), a value that can be interpreted as a distance from a Nash equilibrium (called **exploitability** in the two-player setting). NashConv is easy to compute in small enough games [45]; for CFR's values see Appendix E.1.

**Effect of Exploration and Meta-Strategy Overview**. We now analyze the effect of the various meta-strategies and exploration parameters. In Figure 4, we measure the mean area-under-the-curve (MAUC) of the NashConv values for the last (right-most) 32 values in the NashConv graph, and exploration rate of $\gamma = 0.4$. Figures for the other values of $\gamma$ are in Appendix E, but we found this value of $\gamma$ works best for minimizing NashConv. Also, we found that decoupled replicator dynamics works best, followed by decoupled regret-matching and Exp3. Also, it seems that the higher the level, the lower the resulting NashConv value is, with diminishing improvements. For exploitation, we found that $\gamma = 0.1$ was best, but the meta-solvers seemed to have little effect (see Figure 10.)

**Comparison to Neural Fictitious Self-Play**. We now compare to Neural Fictitious Self-Play (NFSP) [40], an implementation of fictitious play in sequential games using reinforcement learning. Note that NFSP, PSRO, and DCH are all sample-based learning algorithms that use general function approximation, whereas CFR is a tabular method that requires a full game-tree pass per iteration. NashConv graphs are shown for {2,3}-player in Figure 5, and performance vs. fixed bots in Figure 6.

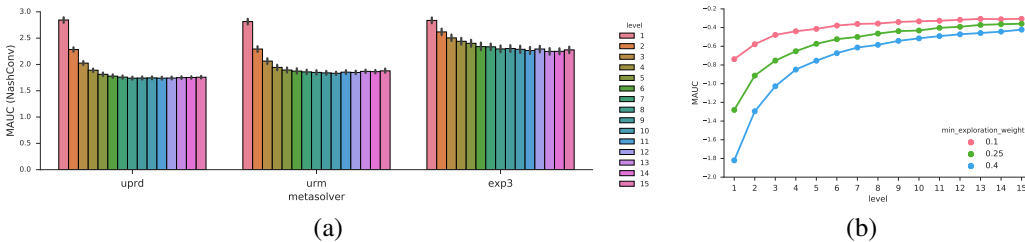

(a)    (b)

Figure 4: (a) Effect of DCH parameters on NashConv in 2 player Leduc Poker. Left: decoupled PRD, Middle: decoupled RM, Right: Exp3, and (b) MAUC of the exploitation graph against cfr500.

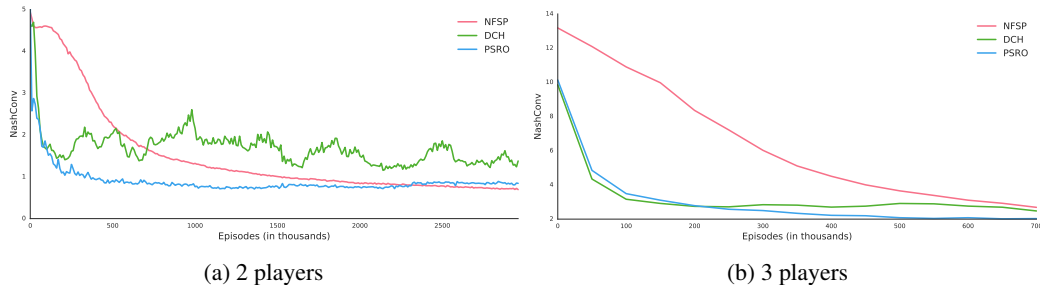

(a) 2 players           (b) 3 players

Figure 5: Exploitability for NFSP x DCH x PSRO.

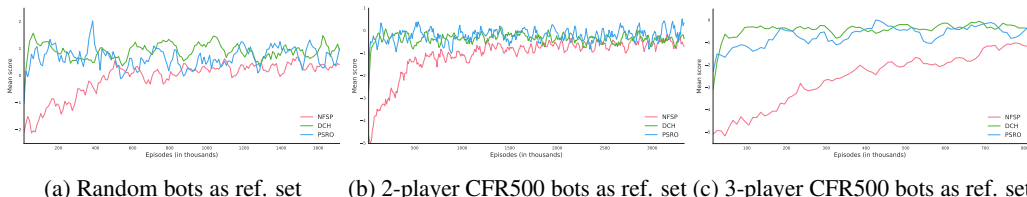

(a) Random bots as ref. set     (b) 2-player CFR500 bots as ref. set (c) 3-player CFR500 bots as ref. set

Figure 6: Evaluation against fixed set of bots. Each data point is an average of the four latest values.

We observe that DCH (and PSRO) converge faster than NFSP at the start of training, possibly due to a better meta-strategy than the uniform random one used in fictitious play. The convergence curves eventually plateau: DCH in two-player is most affected, possibly due to the asynchronous nature of the updates, and NFSP converges to a lower exploitability in later episodes. We believe that this is due to NFSP's ability to learn a more accurate mixed average strategy at states far down in the tree, which is particularly important in poker, whereas DCH and PSRO mix at the top over full policies.

On the other hand, we see that PSRO/DCH are able to achieve higher performance against the fixed players. Presumably, this is because the policies produced by PSRO/DCH are better able to recognize flaws in the weaker opponent's policies, since the oracles are specifically trained for this, and dynamically adapt to the exploitative response during the episode. So, NFSP is computing a safe equilibrium while PSRO/DCH may be trading convergence precision for the ability to adapt to a range of different play observed during training, in this context computing a robust counter-strategy [44, 24].

## 5 Conclusion and Future Work

In this paper, we quantify a severe problem with independent reinforcement learners, joint policy correlation (JPC), that limits the generality of these approaches. We describe a generalized algorithm for multiagent reinforcement learning that subsumes several previous algorithms. In our experiments, we show that PSRO/DCH produces general policies that significantly reduce JPC in partially-observable coordination games, and robust counter-strategies that safely exploit opponents in a common competitive imperfect information game. The generality offered by PSRO/DCH can be seen as a form of "opponent/teammate regularization", and has also been observed recently in practice [66, 5]. We emphasize the game-theoretic foundations of these techniques, which we hope will inspire further investigation into algorithm development for multiagent reinforcement learning.

In future work, we will consider maintaining diversity among oracles via loss penalties based on policy dissimilarity, general response graph topologies, environments such as emergent language games [58] and RTS games [96, 84], and other architectures for prediction of behavior, such as opponent modeling [37] and imagining future states via auxiliary tasks [43]. We would also like to investigate fast online adaptation [1, 21] and the relationship to computational Theory of Mind [106, 4], as well as generalized (transferable) oracles over similar opponent policies using successor features [6].

**Acknowledgments.** We would like to thank DeepMind and Google for providing an excellent research environment that made this work possible. Also, we would like to thank the anonymous reviewers and several people for helpful comments: Johannes Heinrich, Guy Lever, Remi Munos, Joel Z. Leibo, Janusz Marecki, Tom Schaul, Noam Brown, Kevin Waugh, Georg Ostrovski, Sriram Srinivasan, Neil Rabinowitz, and Vicky Holgate.

## Footnotes

[1]Appendices are available in the longer technical report version of the paper, see [55].

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
