[Reviews · NeurIPS 2017]

Reviewer 1



Summary: The paper tries to address the policy overfitting issue of independent RL for multiagent games. The authors proposes a conceptual algorithm for learning meta policies in two-player games by extending existing methods and designs metric to measure the overfitting effect of independent RL. Strengths: The literature review is thorough. A new metric evaluating the effectiveness of policy learning for multiagent games manifests some unexplored properties of multiagent RL learning. Weakness: 1. There are no collaborative games in experiments. It would be interesting to see how the evaluated methods behave in both collaborative and competitive settings. 2. The meta solvers seem to be centralized controllers. The authors should clarify the difference between the meta solvers and the centralized RL where agents share the weights. For instance, Foester et al., Learning to communicate with deep multi-agent reinforcement learning, NIPS 2016. 3. There is not much novelty in the methodology. The proposed meta algorithm is basically a direct extension of existing methods. 4. The proposed metric only works in the case of two players. The authors have not discussed if it can be applied to more players. Initial Evaluation: This paper offers an analysis of the effectiveness of the policy learning by existing approaches with little extension in two player competitive games. However, the authors should clarify the novelty of the proposed approach and other issues raised above. Reproducibility: Appears to be reproducible.

Reviewer 2



The paper proposes a problem with current learning methods multi-agent settings (particularly independent learners setting), which is overfitting to the policies of other agents. A novel method to measure such overfitting is proposed - joint policy correlation. A practical parallel learning algorithm is proposed that significantly reduces overfitting under the proposed measure on gridworld and poker domains. The paper is well-motivated and the exposition is generally clear. The methods are technically sound and I see no obvious flaws. The background material review is extensive and thorough and references are excellent. From reading the paper, a few questions still remain: - it seems that parts of the method rely on all agents sharing the same action space. Would the method be able to generalize to heterogeneous agent populations? - in the details on projected replicator dynamics, it is still not clear how the described quantities are used to update meta-strategy sigma_i. An explicit question would be useful to solidify this. The problem of overfitting to other agents is indeed an important one in multi-agent systems and I am happy to see work attempting to quantify and overcome this issue. I believe this work will be of value to the machine learning community and I would like to see it published at NIPS.

Reviewer 3



Summary: "A Unified Game-Theoretic Approach to Multiagent Reinforcement Learning" presents a novel scalable algorithm that is shown to converge to better behaviours in partially-observable Multi-Agent Reinforcement Learning scenarios compared to previous methods. The paper begins with describing the problem, mainly that training reinforcement learning agents independently (i.e. each agent ignores the behaviours of the other agents and treats them as part of the environment) results in policies which can significantly overfit to only the agent behaviours observed during training time, failing to generalize when later set against new opponent behaviours. The paper then describes its solution, a generalization of the Double Oracle algorithm. The algorithm works using the following process: first, given a set of initial policies for each player, an empirical payoff tensor is created and from that a meta-strategy is learnt for each player which is the mixture over that initial policy set which achieves the highest value. Then each player i in the game is iterated, and a new policy is trained against policies sampled from the meta-strategies of the other agents not equal to i. After training, this new policy is added to player i's set of policies. After all agents have trained a new policy, the new entries within the empirical payoff tensor are computed and the meta-strategies are updated. Finally, the process repeats again, training another policy for each agent and so on. This algorithm, called "Policy-Space Response Oracles" (PSRO), is shown to be a generalization of several previous algorithms including independent RL, fictitious play and double oracle. An extension is then developed, called Deep Cognitive Hierarchies (DCH), which operates only on a tractable hierarchy of asynchronously updated policies instead of having meta-strategies learnt over the entire set of all previous policies. PSRO and DCH are then empirically evaluated on several 2D grid world games and Leduc poker. To identify overfitting "joint policy correlation" matrices are constructed on which the "average proportional reduction" metric is computed. This metric makes clear that independent RL does very poorly at generalizing to new opponent behaviours, even when those opponents were trained under identical conditions. Finally, it is shown that PSRO and DCH converge to close to an approximate Nash Equilibrium much faster than Neural Fictitious Self-Play, a related multi-agent algorithm which also uses deep neural network function approximators. Qualitative Assessment: I think the paper presents an interesting and novel solution to the important problem of learning generalizable behaviours in multi-agent environments. The algorithm developed is also scalable and lends itself to be used efficiently with modern function approximators like deep neural networks. The experimental results clearly demonstrate the problem with the naive method of independent RL, with the development of the "joint policy correlation" matrices showing that independent RL gets far lower reward when set against a novel behaviour. It is interesting to see how brittle RL algorithms are if other agents are treated simply as part of the environment. The connections between PSRO and previous methods such as double oracle and fictitious play are informative and give an intuition on what each method is actually computing in an easily comparable way. The related work cited is extensive and covers a wide variety of methods. While it quickly covers a lot of ground, the writing is still clear and easy to follow. The hyperparameter settings are given in the appendix to help others reproduce the work. I have some clarifying questions: 1) Is there a strict reason for using a hierarchy of policies in DCH? Why restrict levels from only considering oracles in previous levels? I understand there is some intuition in that it resembles Camerer, Ho, & Chong’s cognitive hierarchy model, but I was wondering if there was another reason to impose that constraint. 2) In Figure 5 (a) and (b), it seems like NFSP has not plateaued completely. Does NFSP converge to roughly the same NashConv as PSRO/DCH, or a slightly better solution? It would be informative to see the long term performance (until convergence) of all methods. 3) The DCH algorithm seems to have a large number of hyperparameters, with RL agents and decoupled meta-solvers needing to be trained asynchronously. How sensitive is the algorithm's performance to hyperparameters? Did it require a large hyperparameter search to get it working? Typos: 1) There is a sentence fragment at line 280. 2) At line 266, should the upper-case sigma be an upper-case delta (meta-strategy mixture)?